# Enhanced Growth of Lapine Anterior Cruciate Ligament-Derived Fibroblasts on Scaffolds Embroidered from Poly(l-lactide-*co*-ε-caprolactone) and Polylactic Acid Threads Functionalized by Fluorination and Hexamethylene Diisocyanate Cross-Linked Collagen Foams

**DOI:** 10.3390/ijms21031132

**Published:** 2020-02-08

**Authors:** Clemens Gögele, Judith Hahn, Cindy Elschner, Annette Breier, Michaela Schröpfer, Ina Prade, Michael Meyer, Gundula Schulze-Tanzil

**Affiliations:** 1Institute of Anatomy and Cell Biology, Paracelsus Medical University, Nuremberg and Salzburg, Prof. Ernst Nathan Str. 1, 90419 Nuremberg, Germany; clemens.goegele@pmu.ac.at; 2Department of Biosciences, Paris Lodron University of Salzburg, Hellbrunnerstr. 34, 5020 Salzburg, Austria; 3Leibniz-Institut für Polymerforschung Dresden e. V. (IPF), Dresden, Hohe Straße 6, 01069 Dresden, Germany; hahn-judith@ipfdd.de (J.H.); elschner@ipfdd.de (C.E.); breier@ipfdd.de (A.B.); 4Forschungsinstitut für Leder und Kunststoffbahnen (FILK), Meißner Ring 1-5, 09599 Freiberg, Germany; michaela.schroepfer@filkfreiberg.de (M.S.); ina.prade@filkfreiberg.de (I.P.); michael.meyer@filkfreiberg.de (M.M.)

**Keywords:** ligament tissue engineering, ACL, embroidered scaffold, polylactic acid (PLA), poly(l-lactide-*co*-ε-caprolactone (P(LA-CL)), fluorine, cross-linked collagen foam

## Abstract

Reconstruction of ruptured anterior cruciate ligaments (ACLs) is limited by the availability and donor site morbidity of autografts. Hence, a tissue engineered graft could present an alternative in the future. This study was undertaken to determine the performance of lapine (L) ACL-derived fibroblasts on embroidered poly(l-lactide-*co*-*ε*-caprolactone) (P(LA-CL)) and polylactic acid (PLA) scaffolds in regard to a tissue engineering approach for ACL reconstruction. Surface modifications of P(LA-CL)/PLA by gas-phase fluorination and cross-linking of a collagen foam using either ethylcarbodiimide (EDC) or hexamethylene diisocyanate (HMDI) were tested regarding their influence on cell adhesion, growth and gene expression. The experiments were performed using embroidered P(LA-CL)/PLA scaffolds that were seeded dynamically or statically with LACL-derived fibroblasts. Scaffold cytocompatibility, cell survival, numbers, metabolic activity, ultrastructure and sulfated glycosaminoglycan (sGAG) synthesis were evaluated. Quantitative real-time polymerase chain reaction (QPCR) revealed gene expression of collagen type I (COL1A1), decorin (DCN), tenascin C (TNC), Mohawk (MKX) and tenomodulin (TNMD). All tested scaffolds were highly cytocompatible. A significantly higher cellularity and larger scaffold surface areas colonized by cells were detected in HMDI cross-linked and fluorinated scaffolds compared to those cross-linked with EDC or without any functionalization. By contrast, sGAG synthesis was higher in controls. Despite the fact that the significance level was not reached, gene expressions of ligament extracellular matrix components and differentiation markers were generally higher in fluorinated scaffolds with cross-linked collagen foams. LACL-derived fibroblasts maintained their differentiated phenotype on fluorinated scaffolds supplemented with a HMDI cross-linked collagen foam, making them a promising tool for ACL tissue engineering.

## 1. Introduction 

The anterior cruciate ligament (ACL) is a hypocellular and hypovascular dense connective tissue. It runs intraarticularly, directly connecting the inner surface of the lateral femur condyle with the anterior tibial plateau, thereby ensuring knee stability. Since self-healing of the ruptured ACL is unlikely, a reconstruction must be carried out [1,2]. In order to prevent the development of osteoarthritis as a typical sequela of knee instability, resulting in chronic knee pain, autologous tendon grafts are currently used as a gold standard for ACL reconstruction [3]. However, in light of the very forceful anatomical double-bundle structure of the ACL, these tendon autografts may not be able to permanently withstand the mechanical loads [3]. Numerous experiments with various cell-free biomaterials to support ACL reconstruction have already been carried out, but with little success for clinical applications [4,5,6,7]. Therefore, a tissue engineered ACL reconstruction is still urgently needed. Since the ACL is a hypocellular tissue, a stable and transient cell carrier is required to provide stability until a sufficient extracellular matrix (ECM) is synthesized by the cells and novel ACL tissue is formed.

Biomaterials that are appropriate as transient cell carriers for ACL reconstruction must be cyto- and biocompatible and withstand biomechanical stress [5,8]. Synthetic materials such as poly(l-lactide-*co*-*ε*-caprolactone) (P(LA-CL)) or polylactic acid (PLA) are already established in the medical field [9,10]. P(LA-CL) has been recommended for ACL tissue engineering [11]. The advantage of these materials is that they are cytocompatible [12,13], degradable over a sufficient long period of time as well as stable enough and suitable for the embroidery process [14]. Embroidered structures have promising biomechanical properties and are suited for colonization with lapine ACL (LACL)-derived fibroblasts [13,15]. PLA has been described as a degradable, biomechanically robust, highly biocompatible and well processable polymer [16]. However, a disadvantage is that PLA alone provides only limited cell adhesion, as described for ACL fibroblasts previously [12]. Major influencing factors are its hydrophobicity [16] and the release of acidic degradation products [17]. In contrast to PLA, P(LA-CL) threads are more suitable for cell attachment, as shown in previous works [12]. Nevertheless, functionalization of P(LA-CL) scaffolds has also been recommended [13]. Accordingly, Chen et al. reported that functionalization of P(LA-CL) scaffolds with silk fibroin led to a higher proliferation of a human fibroblast cell line after 7 days [18]. 

For this reason, a scaffold embroidered from both P(LA-CL) and PLA threads has been developed and supplemented with collagen foam. Hoyer et al. [13] showed that a collagen foam integrated into P(LA-CL) scaffolds increased the colonization with LACL-derived fibroblasts, but did not mediate the direct cell-thread contact when incorporated into polydioxanone scaffolds. Moreover, the adherence of the collagen foam to the threads and its stability in the scaffold was limited [13]. From this, it can be concluded that additional functionalization and a special collagen cross-linking technique provide an advantage. 

Collagen is a major structural protein of the extracellular matrix (ECM) of many tissues and supports their resilience as a stabilizing framework. The ECM of the ACL consists of more than 85% of collagen type I and III as well as of 12% sulfated glycosaminoglycans (sGAGs) (related to dry tissue) [19]. Collagen has successfully been used to improve cell adherence on synthetic polymers such as P(LA-CL) [13]. It represents a helpful component to supplement polymer scaffolds for ligament tissue engineering, leading to increased collagen deposition and mechanical properties of the reconstructed ligament in vivo compared to the implanted pure polymer [20]. Cross-linking of collagen is a recommended strategy for its stabilization, but the particular cross-linking agent should be selected properly in order to obtain a robust and cytocompatible construct [21]. 

Chemical cross-linking stabilizes triple helical collagen against enzymatic degradation and denaturation by temperature [22]. Many different reagents have been used in the past such as glutaraldehyde, carbodiimides, isocyanates, epoxides as well as polyvalent metal cations. These chemicals usually react with different side chains of the collagen molecules [22]. This work aims at achieving high cytocompatibility and for this reason hexamethylen isocyanate (HMDI) and ethyl-3(3-dimethylamino)propylcarbodiimide (EDC) were used to stabilize the collagen’s structure. Both have been investigated in the past regarding the stabilization of biomaterials [23,24,25]. It has been reported that chemically cross-linked collagen using EDC facilitates higher rates of human fibroblasts migration and new tissue formation than physical cross-linking based on dehydrothermal treatment or ultraviolet light [26]. A multitude of different material compositions, comprising a collagen component with increased stability against degradation and thus diverse application areas like bone tissue engineering [27], soft tissue engineering [28] and also decellularized scaffolds [29] were treated with HMDI for chemical cross-linking to increase their stability. Therefore, HMDI was also included as a cross-linker in the present study. Additionally, EDC and HMDI cross-linking were particularly suitable for dermal implants [30], for hard tissue application [31] or for bioprosthetic mesh materials [32].

Functionalization can not only be executed by coating with cross-linked collagen but also with a fluorination of the material surface to upgrade the actual basic PLA framework e.g., leading to increased hemocompatibility as an indicator of cytocompatibility [33] or elevated cell adhesion [34]. The effect of gas-phase fluorination on biomaterials is not fully understood but includes radical chain reactions on the material surface, resulting in fluorine radical formation, which opens C–H bonds and generates novel C-F, C-F2 and C-F3 groups ([35], Schroepfer et al.: Gas-phase fluorination on PLA improves cell adhesion and spreading, submitted in 2020 *to ACS Omega*). Based on the results of a previous work (Schroepfer et al.), the embroidered P(LA-CL)/PLA scaffolds in the present study were functionalized with a 10% fluorine concentration during synthetic air flushing and a native purified collagen foam. 

The focus of the present study was to ensure cell adhesion and the differentiated phenotype of LACL-derived fibroblast through a suitable modification of the surface of the bi-component scaffold made of P(LA-CL) and PLA. The hypothesis of the present study is that the adherence, proliferation and migration ability of LACL-derived fibroblasts could potentially be increased in P(LA-CL)/PLA scaffolds treated by gas-phase fluorination and supplemented with a collagen foam. A second assumption was that a higher stability and superior scaffold colonization can be reached by cross-linking and thereby stabilizing of the collagen foam within the scaffold [36]. To prove this, two cross-linking methods were tested.

## 2. Results

### 2.1. Effect of Cross-Linking and Fluorination on Colonization of Three-Dimensional Scaffolds by LACL-Derived Fibroblasts

#### 2.1.1. Cytotoxicity of Scaffold Variants

To assess whether the scaffold functionalization exerted cytotoxic effects on cells, cytotoxicity testing according to the International Standard DIN ISO 10993-5:2009 was performed with the murine L929 fibroblasts as a testing cell line. Since LACL-derived fibroblasts were used for the scaffold seeding experiments, these cells were also included in cytotoxicity testing experiments. The embroidered scaffold variants were either non-functionalized (control) or functionalized with gas-phase fluorination, filled with collagen foam and cross-linked with ethylcarbodiimide (EDC) or hexamethylene diisocyanate (HMDI) (Figure 1). The metabolic activity of the L929 fibroblasts was higher than 75% when treated with extracts of all four scaffold variants. The positive control (treatment with dimethylsulfoxide (DMSO)) was lower than 45% and the negative control (treatment with normal growth medium) was 100% (Figure 2).

#### 2.1.2. LACL-Derived Fibroblast Survival on the Scaffold

Due to the fact that none of the four scaffold functionalization variants was cytotoxic, scaffold colonization by LACL-derived fibroblasts was tested. Two seeding approaches were used: a dynamical seeding procedure with a cell suspension as well as a statical seeding strategy in which self-assembled fibroblast spheroids were placed on the scaffold. Since ligament-derived fibroblasts are mechanosensitive, the dynamical culture model was chosen to provide some mechanical stimuli for the cells. The spheroid-based method allows the directed seeding of distinct scaffold areas with cells and served as a suitable method for observation of the adherence as well as the combined spreading and migratory activity of LACL-derived fibroblasts: Spheroids have to attach to the scaffold mediated through direct cell-scaffold interactions in order to enable cell emigration from the spheroid onto the threads. After 7 and 14 days in both culture systems cell survival on the scaffolds was verified with a vitality assay (Figure 3). The live/dead images showed that LACL-derived fibroblasts colonized all four scaffold types (control, only fluorinated, EDC cross-linking and HMDI cross-linking variations, both combined with fluorination).

The scaffold of the control group, however, presented only few clusters of aggregated cells after seven days of dynamical seeding (Figure 3A). After 14 days, it was found that although some dead cells were present, these were distributed in the majority of the living cells (Figure 3I). The shapes of the cell spheroids placed onto the statically seeded scaffold remained mainly round and convex, with only a few cells migrating from the spheroids into the control scaffolds after 7 days (Figure 3E). After 14 days the spheroids were still round, but more flattened, whereby more single cells had moved out of the spheroid and migrated onto the threads of the scaffold (Figure 3M). Fluorinated scaffolds that were seeded dynamically with the cell suspension showed more cell adhesion compared to the non-fluorinated control after 7 days (Figure 3A,B). The results after 14 days did not significantly differ compared to the respective control (Figure 3I,J). At the same time, the spheroid culture on fluorinated scaffolds revealed that the spheroids were more deeply embedded in the scaffold compared to the non-fluorinated controls (Figure 3E,F). A higher number of emigrated and viable cells was observed on the threads around the spheroids after 14 compared to 7 days (Figure 3N) but with no major differences to the control at 14 days (Figure 3M). In comparison to the control, more cells were distributed on the threads after 7 days of dynamical culture in the EDC cross-linked variant supplemented with collagen foam. Formation of some small cell clusters was visible after 7 days (Figure 3C). After 14 days, the number of cells on the threads was higher and the cell arrangement was denser than after 7 days in the dynamical culture (Figure 3K). The spheroid culture on EDC cross-linked scaffolds showed a large colonized area, with only a few dead cells after 7 days (Figure 3G). After 14 days, the spheroid was still visible with more emigrated cells forming small clusters than after 7 days (Figure 3O). The HMDI cross-linked scaffolds showed an even distribution of LACL-derived fibroblasts on the threads only 7 days after seeding with a cell suspension and dynamical culture (Figure 3D). After 14 days, mainly living LACL-derived fibroblasts were still visible, densely colonizing the scaffolds (Figure 3L). Hence, the cell behavior was similar to that after 7 days. The HMDI cross-linked variant showed more threads colonized with LACL-derived fibroblasts than the other three groups. In this variation, the spheroid-based scaffold colonization showed a spheroid with a high number of living and only a few dead cells in the middle of the spheroid after 7 days (Figure 3H). The spheroid containing the majority of vital cells and only a few dead cells at its surface was completely embedded between the threads after 14 days. The spheroid shape was no longer discernible and the threads around the spheroid were colonized by more cells than the three other scaffold variations (Figure 3P).

#### 2.1.3. Cell Morphology on the Scaffold Functionalization Variants Shown by Scanning Electron Microscopy

Scanning electron microscopy (SEM) imaging of primary LACL-derived fibroblasts on the embroidered scaffolds showed direct cell attachment and spreading on the threads after 14 days of dynamical culture (Figure 4A–D). A close interaction of the cells with the scaffold threads could be seen. Fibroblasts on the scaffold without functionalization showed a convex and spindle-shaped cell body with a small number of filopodia. Cells on the scaffold with fluorine functionalization had a flattened cell shape. The fibroblasts were embedded in a filamentous network of collagen in the EDC cross-linked scaffold. The cells were rounded, had a convex cell body and formed a small cluster. Fibroblasts on the HMDI cross-linked scaffolds were elongated and flattened with a high number of cell-scaffold contacts by focal adhesion points. In addition, elongated filopodia could be observed (Figure 4A–D). 

The spheroid based seeding method was used to visualize spreading and migratory activity of the cells. Spheroid attachment requires statical conditions. It showed how the LACL-derived fibroblasts emigrated from the spheroids and tried to colonize their nearest environment. Especially in the HMDI cross-linked scaffolds, a higher number of emigrated cells could be observed in comparison to the non-functionalized scaffolds, where cells remained convex and oriented parallelly to each other in a helicoidally twist around the thread. It was observed that the filopodia of LACL-derived fibroblasts contained multiple cell adhesion points and were generally longer, more stretched and had a higher tendency to orientate along the thread direction in the HMDI cross-linked scaffold when compared to the other variants (Figure 4E–L).

#### 2.1.4. Areas of Scaffold Functionalization Variants Colonized by Viable Cells 

Scaffold areas colonized by viable cells were calculated for all four scaffold variants seeded dynamically with a LACL cell suspension. It was found that the control scaffold group was least colonized after 7 and 14 days. Only about 3% of the surface area of the control scaffold was occupied by cells measured with the ImageJ software. The colonized scaffold areas of the HMDI cross-linked scaffolds were significantly higher than in the three other scaffold variants with 18% (7 days) and 17% (14 days), respectively (Figure 5A). While not being significant, the fluorine treatment alone was associated with a higher colonization level compared to variants with additional EDC cross-linking and the non-functionalized controls at day 7.

#### 2.1.5. Numbers of Cells Colonizing the Scaffold Functionalization Variants 

Analysis of the cell numbers, estimated from DNA content, was performed with the dynamically seeded scaffold variants. They showed significantly lower cell numbers in the control (with no functionalization) compared to the HMDI cross-linked scaffold after 7 days. The cell content in the non-functionalized controls was the lowest after 7 days in comparison to the other three variants. The cell number per scaffold was around 1.5 × 10^5^ cells in the scaffold with fluorine functionalization alone. Generally, there was a slight and non-significant cell number reduction on each scaffold type after 14 days in comparison to the 7-day time point. The cell content was higher in the HMDI cross-linked scaffold than in the other groups after 14 days (Figure 5B), albeit not significantly.

#### 2.1.6. Metabolic Activity of LACL-Derived Fibroblasts on Scaffold Functionalization Variants

The metabolic activity of the LACL-derived fibroblasts was determined in the dynamically seeded scaffold variants. Although the differences were not significant for the 7 day time point, all functionalized scaffold variants (fluorine, fluorine + collagen + EDC and fluorine + collagen + HMDI) showed a slightly higher metabolic activity compared to the control (without functionalization). After 14 days, the metabolic activity of LACL-derived fibroblasts on the different scaffold types was generally lower than the activity at the 7-day time point (not significant) (Figure 5C). 

#### 2.1.7. sGAGs Synthesized by LACL-Derived Fibroblasts on the Scaffold Variants

sGAG content was analyzed by using the DMMB assay to proof the synthetic activity of LACL-derived fibroblasts within the scaffold variants. LACL-derived fibroblasts produced sGAGs in all scaffold variants cultured in vitro under dynamical conditions for 7 and 14 days. The amount of sGAGs did not significantly differ between the four scaffold types after 7 days. In the fluorine-functionalized scaffold both with and without cross-linked collagen foams, no significant differences were found when comparing the sGAG content of 7 and 14 days (Figure 5D). However, a significant increase in sGAGs could be measured when comparing sGAG synthesis at 7 and 14 days in the non-functionalized scaffolds (control). At day 14, the control contained significantly more sGAGs than all other scaffold functionalization variants at the same time. 

#### 2.1.8. Migration Distance of LACL-Derived Fibroblasts into Scaffold Variants

The extent of cell penetration into inner parts of the scaffold was measured for all four scaffold variants after DAPI staining (Figure 6). Cells were seeded dynamically with a LACL cell suspension and cultured for 7 days. Vertical cross-sections of scaffolds of the control group showed that the LACL-derived fibroblasts were mostly localized at the surface of the scaffold and formed cell clusters inside the scaffold. Therefore, the penetration depth of cells was significantly lesser in the control scaffolds in comparison to the other three variants. After 7 days, the penetration depth of the cells in the HMDI cross-linked scaffolds was significantly larger colonizing more than 45% of the cross-sectional diameter compared to the control groups, solely fluorinated scaffold group and the EDC cross-linked group. It seemed that both outer layers and most of the inner layer of the HMDI cross-linked scaffolds were nearly completely penetrated by LACL-derived fibroblasts (Figure 6D). 

#### 2.1.9. Expression of Ligament-Related Genes in Scaffold Cultures

The expression of ligament-related genes was measured to assess whether the differentiated phenotype of ligament-derived fibroblasts is maintained on the scaffold. Generally, a high inter-donor variance was observed with all genes investigated. Hence, the differences did not reach the significance level. At 24 h, the relative collagen type I (alpha1 chain, COL1A1) expression was the highest in the non-functionalized scaffold in comparison to the other three scaffold groups. The relative gene expression of COL1A1 decreased after 7 days but increased after 14 days again in all four scaffold variants, with no significant differences between them (Figure 7A). Although there was no significant difference to the functionalized variants, the relative decorin (DCN) gene expression of LACL-derived fibroblasts was the highest after 24 h in the scaffolds without functionalization. Over the entire observation, the relative DCN expression decreased in the non-functionalized scaffold but remained stable in the functionalized variants. The relative DCN expression in the scaffold with fluorine functionalization was nearly the same at 7 and 14 days. At 24 h, the relative DCN expression was the lowest in the EDC cross-linked scaffold in comparison to the other three functionalized scaffolds (Figure 7B). At 24 h and at 7 days the relative tenascin C (TNC) expression was the highest in the HMDI cross-linked variant. At 14 days the relative TNC gene expression showed no major difference between the four scaffold variants (Figure 7C). LACL-derived fibroblasts revealed higher transcriptional activity of the Mohawk (MKX) gene in the control group after 24 h in comparison to other groups after 24 h. The relative MKX gene expression slightly increased over time in the HMDI cross-linked variant. After 14 days, the relative MKX gene expression increased non-significantly in the functionalized scaffolds compared to the control group (Figure 7D). In the scaffold with only fluorine functionalization a continuous increase over time as well as the highest relative tenomodulin (TNMD) expression could be observed. In addition, a time-dependent elevation in the relative TNMD gene expression was observed in the EDC cross-linked scaffold (Figure 7E). 

## 3. Discussion

Scaffold based ligament regeneration depends on crucial cellular processes including cell adhesion, proliferation, differentiation and tissue maturation [37]. Scaffolds prepared from synthetic biomaterials that can meet biomechanical requirements [12] often do not sufficiently support these cell needs. Hence, an appropriate functionalization of the material is necessary. This study focused on scaffold functionalization tailored for ACL tissue engineering and characterized LACL-derived fibroblasts performance in a functionalized scaffold in detail. The results of the experiments revealed that scaffold fluorination and supplementation with collagen foam stabilized by HMDI cross-linking are appropriate for ACL scaffold functionalization. 

Moreover, a high cytocompatibility of all scaffold surface modifications, irrespective of whether cross-linked or not, could be proven with both a fibroblast cell line (L929) or primary LACL-derived fibroblasts with no significant differences for HMDI and EDC. The vitality assay showed that the all scaffolds variants could be colonized with primary LACL-derived fibroblasts not only using a dynamical but also with a statical cell spheroid-based method. This underlines the cytocompatibility of the cross-linked collagen foam within the scaffolds. The HMDI cross-linked variant revealed the largest colonized scaffold areas corresponding with the highest cell content calculated from the DNA content. However, there was no increase in the colonized area from 7 to 14 days in none of the four scaffold variants. Moreover, the cell number decreased slightly but not significantly on all variants after 14 days in culture. This allows the assumption that impaired cellularity could mimic the natural switch from cell proliferation to matrix synthesis, organization and differentiation in maturing ligaments. Generally, mature ligaments possess a lower cell content than immature tissues [38], but decreased cell numbers could also be caused by a loss of cell adhesion. 

It has been shown previously that surface modifications by fluorination using xenondifluoride as a fluorination agent increased mesenchymal stromal cell adhesion on graphene sheets and their neuronal differentiation [39], but the effects of fluorine or of material surface alterations mediated by fluorination on LACL-derived fibroblasts are still unknown. Therefore, effects of gas-phase fluorination on cells are currently studied in more detail ([29] *Schroepfer* et al.: Gas-phase fluorination on PLA improves cell adhesion and spreading, submitted 2020 *to ACS Omega*). The stimulation of various G proteins through aluminofluoride [40] was described. *Wardas* and colleagues gave evidence that fluorine leads to an activation of the lysyl and prolyl hydroxylases, which are essential for collagen production, inducing a slight increase in collagen synthesis [41]. An impaired fibroblast growth has also been reported in the presence of fluorine concentrations higher than 0.2 parts per million (ppm) [42]. While it remains questionable whether cells indeed take up fluorine, it is necessary to consider the possibility that only indirect effects on cells occur, mediated by fluorination-induced changes e.g., of the surface charges and hydrophilicity of a respective material treated. The fluorinated scaffolds revealed only a slight but not significant stimulatory effect on LACL-derived fibroblast growth and activity on the scaffolds, observed after 7 days, but the degree of cell penetration into the scaffolds was indeed significantly higher than in the control scaffolds at the same time point suggesting an influence of fluorine on cell performance. 

Compared with the fluorination alone, the stimulatory influence on cell colonization of deeper scaffold areas exerted by a collagen foam was significantly higher either cross-linked by EDC or more pronounced, when treated with HMDI. One could hypothesize that the refibrillated collagen presents natural binding motifs for integrin mediated cell adhesion and migration. The measured cell penetration at day 7 of around 45% of the inner scaffold diameter for the HMDI cross-linked variant suggests that the porosity of the scaffolds might be within a suitable range for ACL-derived fibroblasts. The porosity was calculated previously to be 65%–75% for the non-functionalized scaffolds [43]. The retention of cells in the scaffold over the entire observation time suggests also sufficient stability of the collagen foam by both cross-linking techniques. 

In this study, the statical spheroid seeding approach was used as a simple and versatile model to study cell adherence and migration onto the scaffold threads. Spheroids present a tool for targeted and directed colonization of defined scaffold areas by the desired cell type. Hence, a spheroid-based seeding has become a well-established method in regenerative medicine, not only with stem cells [44], endothelial cells [45] or osteoblasts [46] but also with ligament- and particularly ACL-derived fibroblasts, as previously shown [15,38]. The results after 14 days indicate that long-term cultivation is possible with both the dynamical and statical (spheroid-based) seeding approaches. Scaffolds with no functionalization (control group) showed lesser numbers of emigrated cells and smaller spheroid sizes, suggesting lower cell-thread interaction and thereby spreading than the three scaffold types with functionalization. SEM analysis of the spheroids allowed a detailed insight into spheroid attachment on the scaffold variants as well as the emigration of the fibroblasts compared to live/dead staining. The more flattened spheroid and cells on the HMDI cross-linked variant suggest a more intense cell-thread interaction.

The parallel cell arrangement and longitudinal orientation according to the thread direction observed in the present study by SEM and live/dead staining is comparable to cell alignment in the native ACL [47] and was observed in another scaffold type consisting of braided nanofibers [48]. 

In addition to cell distribution, synthetic activity is important for tissue formation. The present study revealed that primary LACL-derived fibroblasts are still highly metabolically active in all investigated scaffold variants with no significant difference between 7 and 14 days. If the synthetic capacity is considered, a comparatively low sGAG content was observed at 7 days of dynamical cultivation, which is not unusual for ligament-derived fibroblasts compared to chondrocytes [49,50]. However, it increased in the non-functionalized control group, becoming significantly higher than in all the other scaffold variants after 14 days. The most important proteoglycan containing sGAGs in ligaments and tendons is the small leucine-rich proteoglycan named decorin (DCN). It was investigated here at the gene expression level. In general, a functionalization led to a higher expression after 14 days in comparison to the non-functionalized scaffold group after 14 days. DCN is involved in the regulation of collagen fibrillogenesis, modulation of growth factor activity and regulation of cellular growth [51,52]. DCN has a life-long expression [53] and it is involved in tendon healing [54]. Therefore, the decrease in relative DCN gene expression in the HMDI cross-linked scaffold after 7 days could be part of a natural remodeling process in the tissue-engineered construct. Since the particular effect of EDC and HMDI on ligamentogenesis by LACL-derived fibroblasts in three-dimensional (3D) scaffolds has not been investigated so far, gene expression analysis of other ligament-related genes was performed. 

Collagen type I is the most abundant protein in the extracellular matrix of ACL. An increase in the COL1A1 gene expression level as observed at 14 days compared to 7 days in the present study is in accordance with another study [55] and could indicate the starting point of tissue formation. 

Gene expression of the fibroblast marker and extracellular glycoprotein tenascin C (TNC) typically observed in ACL-derived fibroblasts [13] could be confirmed, showing no significant difference between functionalization variants. TNC gene expression after 7 days, especially in the HMDI cross-linked scaffold, confirmed our hypothesis that collagen cross-linking offers LACL-derived fibroblasts a more suitable opportunity for cell matrix formation than other functionalized scaffolds. 

Expression of the tendon/ligament-specific transcription factor Mohawk (MKX) is influenced by surface topology as shown for other synthetic polymer scaffolds including braided structures and mesenchymal stromal cells undergoing tenogenesis [56]. It is also known to be regulated by mechanical forces and to mediate mechanoresponses [57,58]. For this reason, we used a dynamical culture approach for gene expression analysis in the present study. The MKX gene expression of LACL-derived fibroblasts increased non-significantly after 14 days compared to 7 days, suggesting the maintenance of the ligament-specific phenotype in the scaffolds [59]. Interestingly, no significant differences in MKX expression between the non-functionalized and functionalized scaffolds were observed during the entire observation time. Although MKX is known to regulate type I collagen gene expression [59], no correlations could be found at the gene expression level. 

Tenomodulin (TNMD) is a type II transmembrane glycoprotein of tendons and ligaments, which participates in tendon development, proliferation and maturation and hence, is designated as tendon marker [60]. The relative expression of TNMD in the four different scaffold variation increased after 14 days in culture, especially the functionalized scaffolds with fluorine and EDC cross-linking showed high expression after 14 days compared to the other groups [15,61].

In summary, a new scaffold functionalization strategy based on a fluorination and a cross-linked refibrillated collagen foam has been successfully developed and evaluated for ACL tissue engineering using a lapine primary ACL cell model as basis for future research in a rabbit ACL reconstruction approach. Functionalization of the scaffold in itself supported cell adherence, cell content and the differentiated ACL phenotype more extensively than without functionalization. This study focused solely on cytocompatibility and the cell response to scaffold functionalization. Cell retention over the whole observation time suggested sufficient stability of the cross-linked collagen foam. Future studies using only a simple dynamical culture procedure should provide stronger evidence how the ligamentogenic cellular response of the LACL-derived fibroblasts (cell orientation, proliferation and specific extracellular matrix synthesis) can be optimized by uniaxial cyclic tension of the functionalized embroidered P(LA-CL)/PLA scaffolds. 

## 4. Materials and Methods 

A schematic overview over the experimental design and the four scaffold variants functionalized with gas-phase fluorination and supplemented with cross-linked collagen foam is given in Figure 1. 

### 4.1. Preparation of P(LA-CL)/PLA Scaffolds

Two different thread materials were used for scaffold preparation. A monofilament suture thread made of P(LA-CL) (USP 7-0, Gunze Ldt., Osaka, Japan) and a melt spun multifilament consisting of six filaments made of PLA (*T*t = 155 dtex, Ingeo biopolymer 6202D (NatureWorks, Minnetonka, MN, USA), melt spun at IPF Dresden, Dresden, Germany) were processed using embroidery technology (JCZ 0209-550, ZSK Stickmaschinen GmbH, Krefeld, Germany) [43,62]. P(LA-CL) was used as upper and PLA as lower thread during the embroidery process. Embroidered scaffolds were composed of three plies with a zig-zag pattern design (1.8 mm stitch length, 15° stitch angle and 0.2 mm duplication shift). In an additional manufacturing step, these three plies were stacked and locked together to get a 3D scaffold. Pattern design and composition were chosen according to previous experiments regarding to the mechanical properties [12,62]. The fabrication of the scaffolds was performed on a water-soluble non-woven made of polyvinyl alcohol (PVA, Freudenberg Einlagestoffe KG, Weinheim, Germany). The fabric was washed out three times for 30 min in water on a compact shaker (KS 15 A, Edmund Bühler GmbH, Bodelshausen, Germany). After that, the remaining porous scaffolds were dried at room temperature (RT). The porosity of the scaffolds was determined by µCT as described in [43] and is in a range of 65%–75% (continuously through the whole structure). The control scaffold with no functionalization was then directly used in the cell culture.

### 4.2. Functionalization of the Scaffolds

First, the embroidered scaffolds were fluorinated. This was done at the FILK (Freiberg, Germany) in a fluorination batch reactor (Fluor-Technik-System GmbH, Lauterbach, Germany) in a mixture of 10% fluorine gas in air for 60 s. After the fluorination process, the two scaffold variants with collagen, were flushed with synthetic air (see above) and then functionalized with a solution of bovine acid soluble collagen refibrillated with phosphate buffer and NaCl. The resulting hydrogel within and around the scaffold was desalted and lyophilized to form a foam between the threads of the embroided polymer.

The collagen cross-linking was performed with the gas-phase of HMDI in an exsiccator or using 2% EDC dissolved in pure ethanol. Due to the different functionalization process, a loss of the violet thread color in the EDC cross-linked scaffold was seen, as shown in in Figure 1E3. 

### 4.3. Isolation of Fibroblasts from Lapine ACLs 

LACL-derived fibroblasts were isolated from 7 female, healthy ACLs of New Zealand Rabbits (approximate 12 months) derived from the abattoir. Explanted LACLs were sliced into 2 mm^2^ pieces and placed into a T25 culture flask with growth medium for several weeks (Dulbecco’s Modified Eagle´s Medium (DMEM)/Ham’s F12 medium (1:1, Merck KGaA, Darmstadt, Germany) supplemented with 10% fetal bovine serum (FBS, Merck KGaA), 1% penicillin/streptomycin solution (Merck KGaA), 25 μg/mL ascorbic acid (Sigma-Aldrich, Munich, Germany), 2.5 μg/mL amphotericin B (Merck KGaA), MEM amino acid solution (Sigma-Aldrich). After 7–10 days, emigrating cells could be harvested using 0.05% trypsin/0.02% EDTA treatment and be further expanded. 

### 4.4. Cytotoxicity Test

The murine fibroblast cell line L929 was used for biological evaluation of cytotoxicity according to the international standard ISO 10993-5:2009. Thawed primary LACL-derived fibroblasts (three different donors, all derived from the passage 1) and L929 cells were seeded with an initial density of 1.0 × 10^4^ cells/cm^2^ in cell culture flasks and cultured in growth medium until 80%–90% confluence at 37 °C and 5% CO_2_ was reached. Growth medium was changed three times a week. Extracts were prepared to determine potential cytotoxic effects of the four scaffold types. Growth medium was used as extraction medium. The 30 mm long scaffolds (1 mm thickness, 4 mm width) were sterilized with 70% ethanol for 2 h. Scaffolds were washed three times 10 min with sterile aqua dest. to get rid of the ethanol. Sterile scaffolds were incubated in 2 mL extraction medium at 37 °C and 5% CO_2_ for 48 h under aseptic conditions using sterile, chemically inert cell culture plates. Fibroblasts were seeded in 96-well cell culture plates with an initial density of 1 × 10^4^ cells per cm^2^ and the cells were incubated for 24 h at 37 °C and 5% CO_2_ to allow cell adherence. The medium was removed and the cells were cultured with 100 µL extract or control solutions per each well for 24 h at 37 °C and 5% CO_2_. A 10% dimethyl sulfoxide solution (DMSO, Carl Roth GmbH, Karlsruhe, Germany) was diluted in growth medium andused as a positive control. Pure growth medium was applied as negative control. After 24 h of incubation with the respective extracts, the medium was completely discarded. 80 µL growth medium and 20 µL [3-(4,5-dimethylthiazol-2-yl)-5-(3-carboxymethoxyphenyl)-2-(4-sulfophenyl)-2H-tetrazolium, inner salt; MTS] solution (CellTiter 96® Aqueous One Solution Cell Proliferation Assay, Promega GmbH, Walldorf, Germany) were added to each well. The cells were incubated for an additional 2 h under standard culture conditions and absorbance was measured photometrically at a wavelength of 490 nm (Tecan Austria GmbH, Grödig, Austria). 

### 4.5. Determination of the Effect of Different Functionalized Scaffold Variants

#### 4.5.1. Scaffold Colonization

Four differently functionalized scaffold variants were seeded using two different strategies (dynamical and spheroid based statical seeding) to determine a suitable collagen cross-linking method in the embroidered 3D ACL scaffold. 

##### Dynamical Scaffold Culture

Before colonization, the sterilized scaffolds were pre-incubated for one hour in FBS. Scaffolds were seeded under rotatory conditions with a suspension of 8333,3 LACL-derived fibroblasts/mm^3^ scaffold in 5 mL growth medium in TubeSpin bioreactor tubes (TPP, Trasadingen, Switzerland) using a rotator device (Bartelt GmbH, Graz, Austria) with 36 rpm at 37 °C. Medium changes were executed every 2 days until the cultivation was stopped at day 7 and 14. Nine independent experiments with cells derived from nine different donors were performed. 

##### Spheroid Based Statical Scaffold Colonization

LACL-derived fibroblasts were cultured until a cell density of 90% confluence was reached. Cells were detached with 0.05% typsin/0.02% EDTA (Merck KGaA, Darmstadt, Germany) and counted with a hemocytometer. Spheroids were achieved by using the hanging drop method as described previously [47] with a cell number of 2.5 × 10^4^ per spheroid. After two days of spheroid formation at 37 °C and 5% CO_2_, they were carefully harvested. Ethanol-sterilized, PBS washed and FBS preincubated (for 1 h) scaffolds were colonized with 10 spheroids per scaffold and cultured statically for 7 and 14 days in growth medium. Complete medium changes were performed every second day. Three independent experiments with cells derived from three different donors were performed. 

#### 4.5.2. Cell Survival

Live/dead staining using 1 µL propidium iodide (PI, 1% stock solution, Thermo Fisher Scientific, Darmstadt, Germany) and 5 µL fluorescein diacetate (FDA, stock solution: 3 mg mL^−1^ in acetone, Sigma-Aldrich) in 1 mL PBS was performed to examine the vitality of the fibroblasts on the scaffolds after 7 and 14 days. For performing the live/dead staining, the scaffolds were removed from growth medium. Then, 50 µL of stain solution was added and transferred to a microscopic cover slide. After a 5 min incubation period at RT, the fluorescence of live and dead cells was monitored using a Leica TC SPEII confocal laser scanning microscope (Leica, Wetzlar, Germany). Based on the pictures with only vital cells, the colonized area was measured with the ImageJ program. Three independent experiments were performed with three different observation microscopic fields by the CLSM. In summary, nine microscopic fields of each scaffold variant were analyzed and the measured values were related to the whole picture size.

#### 4.5.3. Measurement of the Penetration Depth of Cells into the Scaffolds

The cell nuclei stained with 4′,6′-diamidino-2-phenylindol (DAPI) were assessed using confocal laser scanning microscopy (CLSM). Scaffolds were rinsed with Tris buffered saline (TBS: 0.05 M Tris, 0.015 NaCl, pH 7.6) before being incubated with protease-free donkey serum (5% diluted in TBS with 0.1% Triton X100) for cell permeabilization for 20 min at RT. Cell nuclei were stained using DAPI (Roche, Mannheim, Germany) for 20 min at RT. Subsequently, scaffolds were washed several times with TBS before examined by using CLSM. The penetration depth of vertically cross-sectioned scaffolds was measured with the Leica X 3D image simulation program. Measurement was done starting at the scaffold border extending to the position of the innermost nuclei detected in the scaffold. The relative values were calculated according to the absolute scaffold thickness (cross-section). Three independent experiments were performed. Three pictures of each scaffold were taken. In summary, nine images of each scaffold variant with distance measurements at five different positions in each image were included.

#### 4.5.4. Scanning Electron Microscopy

Scaffolds were primary fixed in 2% paraformaldehyde (PFA), 2.5% glutaraldehyde (Carl Roth, GmbH) in buffer solution (PBS) overnight at 4 °C, washed in PBS four times, each 15 min and secondary fixed in 1% osmiumtetroxide (OsO_4_). Afterwards, the scaffolds were gently rinsed four times, each 15 min in PBS and dehydrated in an ascending ethanol (ETOH) series (70%, 80%, 90% and 96% ETOH, each for 30 min) and three times 100% ETOH for 15 min. Then, the samples were critical point dried, mounted onto specimen holders and sputtered (agar auto carbon coater, Agar scientific Ltd, Essex, UK) with a thin layer of carbon before photos were taken by the SEM (Cambridge Stereoscan 250; ESEM XL30, Philips, Amsterdam, Netherlands) at an accelerating voltage of 20 kV. 

#### 4.5.5. The CellTiter-Blue® Cell Viability Assay

The metabolic activity of the LACL-derived fibroblasts on a quarter (7.5 mm × 4 mm × 1 mm) of a scaffold variant was analyzed after 7 and 14 days of cultivation. 100 µL of the cultivation medium per well were mixed with 25 µL of Alamar blue solution, pipetted onto the scaffold segment and incubated for an additional 6 h. The fluorescence of each sample was measured in triplicates at 560 excitation/590 emission nm in a fluorometric plate reader (Infinite M200 Pro, Tecan, Groedig, Austria).

#### 4.5.6. Quantitative Assays for DNA- and Sulfated Glycosaminoglycan (sGAG) Quantification

By CyQUANT® NF Cell Proliferation Assay the influence of respective treatment on cell division was examined after 7 and 14 days. The standard curve was generated by serial dilution of calf thymus DNA stock solution (1 mg mL^−1^) with TRIS/EDTA (TE)-buffer (10 mM TRIS (pH 8.0), 1 mM EDTA in Aqua dest). For the standard curve, 25 µL of the serial calf thymus DNA dilutions was mixed with 25 µL of CyQuant dye solution (Hank`s balanced salt solution (HBSS) + dye binding solution 1:250 (ThermoFisher Scientific Inc., Waltham, MA, USA)). After analyzing time points (7 and 14 days) of 3D culture, scaffolds had to be transferred to RNase and DNase using free 2 mL safe seal tubes (Sarstedt AG & Co. KG, Nürnbrecht, Germany) with 50 µL of the Proteinase K digestion buffer (50 mM Tris/HCl, 1 mM EDTA, 0.5% Tween 20) containing 0.5 mg mL^−1^ proteinase K (PanReac, ApplyChem, Darmstadt, Germany). Samples were homogenized with a 7 mm stainless steel bead (RNase and DNase free, sterile, Qiagen, Hilden, Germany) by using the Tissue Lyser of Qiagen (50 Hz, 5 min, RT). Then, 250 µL of the Proteinase K digestion buffer (50 mM Tris/HCl, 1 mM EDTA, 0.5% Tween 20) containing 0.5 mg mL^−1^ proteinase K (PanReac, ApplyChem) were added. All samples were digested for 16 h at 56 °C under continuous shaking. The enzymatic reaction was stopped by freezing the samples at −20 °C for 30 min. Before DNA quantification, all samples were thawed and then centrifuged for 15 min at 10000 rounds per minutes (rpm). 10 µL of each sample were added to 150 µL TE buffer and thoroughly mixed. Samples were transferred in triplicate with 25 µL of the sample dilution to a black, flat bottom 96-well plate (Brand GmbH, Wertheim, Germany) and mixed with 25 µL of the dye solution (HBSS + dye solution 1:250). Subsequently, plates were covered to be protected from light and incubated at 37 °C for 60 min. The fluorescence of each well was measured in triplicate at 485 excitation/530 emission nm in a fluorometric plate reader. Three independent experiments with cells derived from three different donors were performed. 

For the dimethyl methylene Blue (DMMB) Assay, the same supernatant was used as in the CyQuant Assay. After adequate sample dilution the DMMB solution (ApplyChem) was added (40 mM glycine (Sigma-Aldrich), 40 mM NaCl (Carl Roth GmbH) at pH 3 and DMMB (8.9 mM in ethanol)). Chondroitin sulfate (Sigma-Aldrich) was used as standard. The absorption shift was measured at wave length of λ = 633 nm to λ = 552 nm using a Genios spectral photometer (Tecan, Groedig, Austria). Three independent experiments with cells derived from three different donors were performed. 

#### 4.5.7. RNA Isolation

Colonized scaffolds (all four variants) were snap-frozen after 24 h, 7 days and 14 days (each *n* = 5) and homogenized in RLT-buffer (Qiagen, Hilden, Germany) with a tissue lyser (Qiagen) for 2 times each 3 min at 50 Hz. RNA was isolated and purified using the RNeasy Mini kit according to the manufacturer´s instructions (Qiagen), including on-column DNAse treatment. Purity and quantity of the RNA samples were monitored for RNA content and purity (260/280 absorbance ratio) using the Nanodrop ND-1000 spectrophotometer (Peqlab, Biotechnologie GmbH, Erlangen, Germany). 

#### 4.5.8. Quantitative Real-Time PCR

For cDNA synthesis 120 ng of total RNA were reverse transcribed using the QuantiTect Reverse Transcription Kit (Qiagen) according to the supplier manual. A total of 20 ng cDNA were used for each quantitative real-time PCR (qRT-PCR) reaction using TaqMan Gene Expression Assays (Life Technologies) with primer pairs for type I collagen (*COL1A1*, Oc03396073_g1), decorin (*DCN*, Hs00370384_m1), tenascin C (*TNC*, Oc06726696_m1), Mohawk (*MKX*, Oc06754037_m1), tenomodulin (*TNMD*, Oc03399505_m1 (synonymous: myodulin)) and the housekeeping gene glyceraldehyde 3-phosphate dehydrogenase (*GAPDH*, Oc03823402_g1) as a reference gene (Table 1). qRT-PCR was performed using the real time PCR detector StepOnePlus (Applied Bioscience (ABI), Foster City, USA) thermocycler with the program StepOnePlus software 2.3 (ABI). The relative expression of the gene of interest by the cells on the scaffolds was normalized to the GAPDH expression and calculated for each sample using the ∆∆CT method as described by reference [63]. 

### 4.6. Statistics

Based on the vitality images, the statistical analysis of colonized area on the scaffolds was calculated with the ImageJ1.48v software. All values were expressed as the mean with the standard deviation using GraphPad Prism 8 (GraphPad Software Inc., San Diego, CA, USA). Before testing the normal distribution of the results, the ROUT method of identifying outliers was applied. The normal distribution of the results was analyzed using the Shapiro Wilk test as well as the D’Agostino and Pearson normality test. When the normality test failed, the Kruskal–Wallis one-way analysis of variance on ranks followed by Dunnett T3 post hoc multiple comparisons was performed to evaluate the significance of the performed colonized scaffold surface, the cell number per scaffold, the metabolic activity and the sGAG content per scaffold. Statistical significance was set at a *p*-value ≤ 0.05 (*) and *p*-value ≤ 0.0001 (****). Three or five (gene expression) independent experiments with cells derived from three or five different donors were performed. 

## 5. Conclusions

This study showed that 3D embroidered scaffolds can be functionalized with a gas-phase fluorination and a cross-linked collagen foam. A detailed investigation (cell survival, scaffold colonization, cell numbers, ultra-morphology) revealed that the functionalization strategy using fluorination and cross-linking of the collagen foam by using hexamethylene diisocyante (HMDI)) allowed a higher cell colonization and cell penetration into inner parts of embroidered PLA/P(LA-CL) scaffolds compared to non-functionalized controls. Moreover, it also guaranteed the maintenance of the differentiated ligamentocyte phenotype of ACL-derived fibroblasts over the entire observation time. 

## Figures and Tables

**Figure 1 ijms-21-01132-f001:**
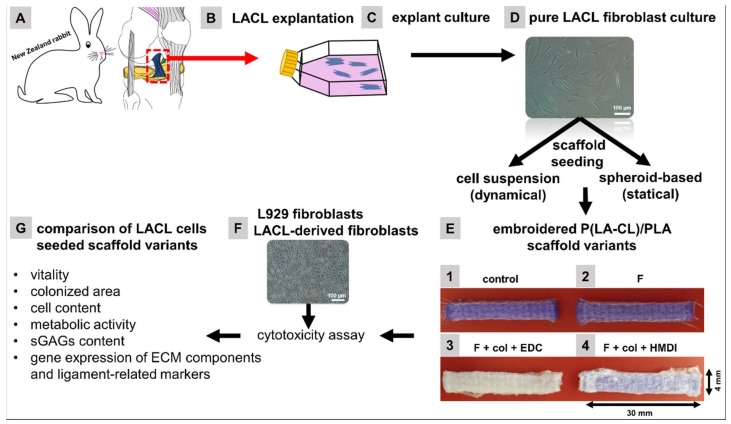
Dissection of the lapine (L) anterior cruciate ligament (ACL) of New Zealand rabbit (**A**). Cultivation of LACL fragments in T25 flasks as an explant culture for cell isolation (**B**,**C**). Expansion of LACL-derived fibroblasts (**D**). Embroidered P(LA-CL)/PLA scaffolds were treated with different functionalization methods (none [**E1**], with 10% fluorine in the gas-phase of the fluorination batch reactor [**E2**], with fluorine + collagen foam + ethylcarbodiimide [EDC] [**E3**] and with fluorine + collagen foam + hexamethylene diisocyanate [HMDI] [**E4**]), tested for cytotoxicity with L929 and LACL-derived fibroblasts (**F**) and then seeded either with a LACL-derived fibroblast suspension in a dynamical or with spheroids in a statiical approach. Downstream analysis (**G**) comprised determination of cell vitality, areas colonized by cells, cell number, metabolic activity, sulfated glycosaminoglycan (sGAG) contents and expression analysis for ligament main ECM and marker genes. **E3**: The EDC treatment evoked a loss of the blue color of the scaffolds. Col, collagen foam; F, fluorinated; EDC, ethylcarbodiimide cross-linked; HMDI, hexamethylene diisocyanate cross-linked.

**Figure 2 ijms-21-01132-f002:**
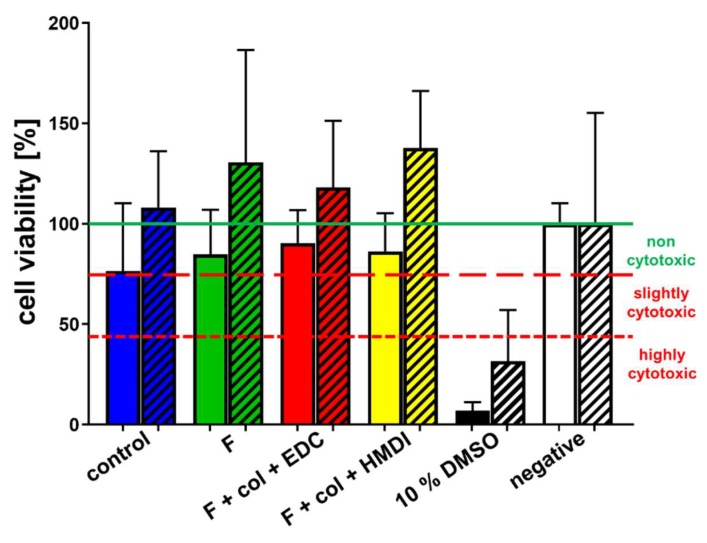
Cytotoxicity Assay of four different scaffold functionalization variants. Viability of L929 (full) and LACL-derived (shaded) fibroblasts treated for 24 h with scaffold extracts (48 h extraction) prepared from the control scaffolds without functionalization (blue), the only fluorinated scaffolds (green) or variants fluorinated + collagen foam + EDC cross-linking (red) or fluorinated + collagen foam + HMDI cross-linking (yellow) was above 70% in all cases. Cell viability of the positive control with 10% dimethylsulfoxide [DMSO, black] was below 45% (highly cytotoxic) and the cell viability of the negative control with culture medium (white) was nearly 100%. The CellTiter 96® Aqueous One Solution Cell Proliferation Assay was used to assess cytotoxicity. Col, collagen foam; F, fluorinated; EDC, ethylcarbodiimide cross-linked; HMDI, hexamethylene diisocyanate cross-linked.

**Figure 3 ijms-21-01132-f003:**
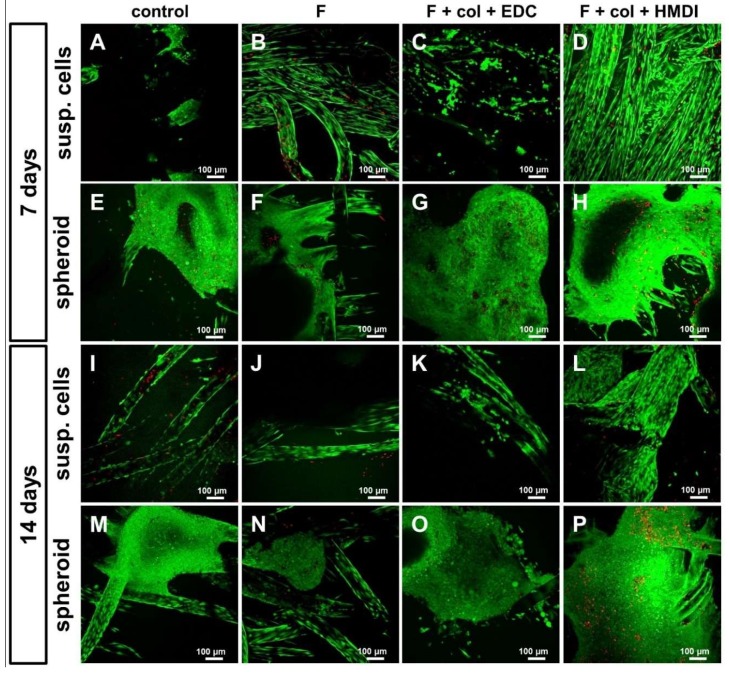
Vitality assay of functionalized scaffold variants cultured either dynamically seeded with suspended cells (susp. cells, **A**–**D**, **I**–**L**) or statically with spheroids (**E**–**H**,**M–P**). Control (**A**,**E**,**I**,**M**), fluorinated (**B**,**F**,**J**,**N**), fluorinated + collagen + EDC (**C**,**G**,**K**,**O**) and fluorinated + collagen + HMDI (**D**,**H**,**L**,**P**) after 7 (**A–H**) and 14 days (**I**–**P**) in culture. Representative pictures of three independent experiments using cells from three different donors show living cells: green and dead cells: red. Scale bars of 100 µm. Col, collagen foam; F, fluorinated; EDC, ethylcarbodiimide cross-linked; HMDI, hexamethylene diisocyanate cross-linked.

**Figure 4 ijms-21-01132-f004:**
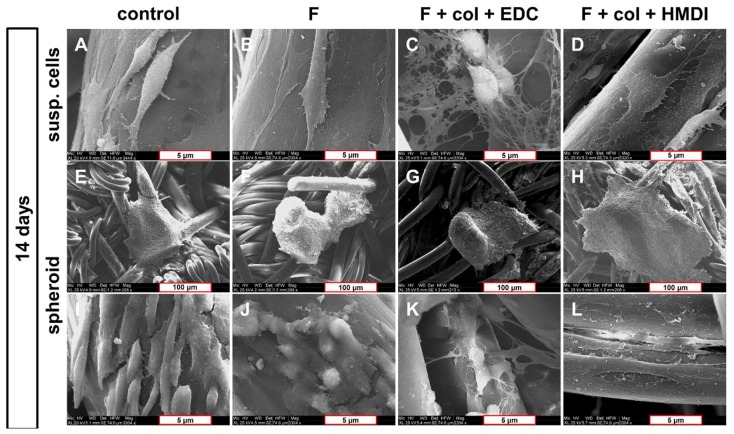
Cell morphology of the LACL-derived fibroblasts on embroidered scaffold functionalization variants using SEM after 14 days in culture. Scaffolds were either dynamically seeded with suspended cells (susp. cells, **A**–**D**) or statically with spheroids (**E**–**L**). Differences in cell shape and distribution can be seen in the dynamical (**A–D**) as well as in the statical culture (**E–L**) on the non-functionalized scaffold (**A**,**E**,**I**), the one with only the fluorination (**B**,**F**,**J**), the fluorinated scaffolds with EDC (**C**,**G**,**K**) and the HMDI cross-linked collagen foam (**D**,**H**,**L**). **A–D** and **I**−**L**: scale bars of 5 µm and **E–H**: scale bars of 100 µm. Col, collagen foam; F, fluorinated; EDC, ethylcarbodiimide cross-linked; HMDI, hexamethylene diisocyanate cross-linked.

**Figure 5 ijms-21-01132-f005:**
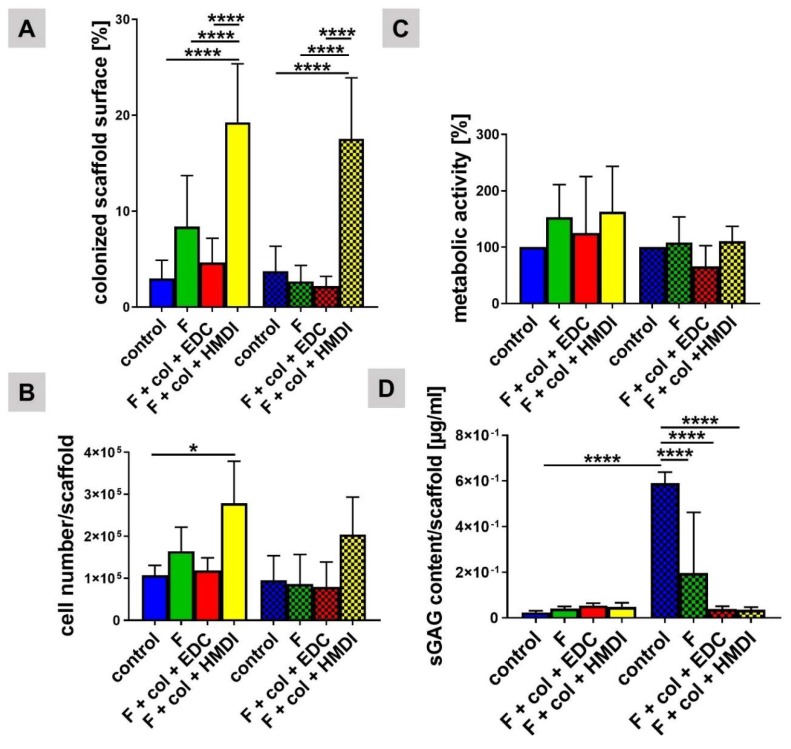
Effect of fluorination and different collagen cross-linking methods on scaffold colonization, metabolic activity, cell numbers and sGAG synthesis of dynamically cultured LACL-derived fibroblasts. Results of colonized scaffold surface calculation (**A**), *n* = 9 independent experiments with LACL-derived fibroblasts of different donors, results of CyQuant Assay (**B**), *n* = 3 independent experiments with LACL-derived fibroblasts of different donors, Alamar Blue Assay to assess metabolic activity (**C**) and dimethyl methylene blue (DMMB) Assay (**D**) after 7 days (fully marked) and 14 days (check marked), *n* = 3 independent experiments with LACL-derived fibroblasts of different donors. One sample *t* test, two-tailed (comparison of different concentrations with control), one-way ANOVA (post hoc Tukey Test) for comparison between the groups. *p* values: * <0.05, **** <0.0001. Col, collagen foam; F, fluorinated; EDC, ethylcarbodiimide cross-linked; HMDI, hexamethylene diisocyanate cross-linked.

**Figure 6 ijms-21-01132-f006:**
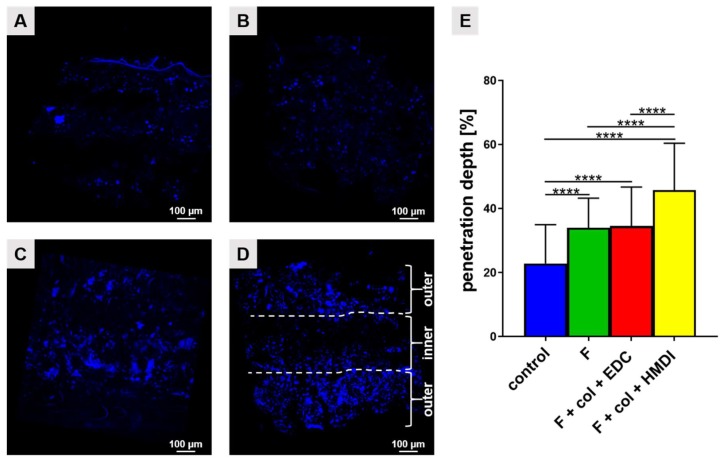
Penetration depth of LACL-derived fibroblasts in functionalized scaffold variants cultured dynamically with suspended cells after 7 days. 4′,6′-diamidino-2-phenylindol (DAPI, blue) staining of cell nuclei in non-functionalized scaffold (**A**), the fluorinated scaffold (**B**), fluorinated + collagen + EDC scaffold (**C**) and fluorinated + collagen + HMDI scaffold (**D**). Representative images of the vertical cross section of scaffolds of three independent experiments using cells from three different donors. Cell nuclei are shown in blue. The three layers of the scaffold were marked with dashed white lines in **D**. Scale bars of 100 µm. The mean of the migration distance of cells into the scaffold is shown (**E**). One sample *t* test, two-tailed (comparison of different concentrations with control), one-way ANOVA (post hoc Tukey Test) for comparison between the groups. p values: **** <0.0001. Col, collagen foam; F, fluorinated; EDC, ethylcarbodiimide cross-linked; HMDI, hexamethylene diisocyanate cross-linked.

**Figure 7 ijms-21-01132-f007:**
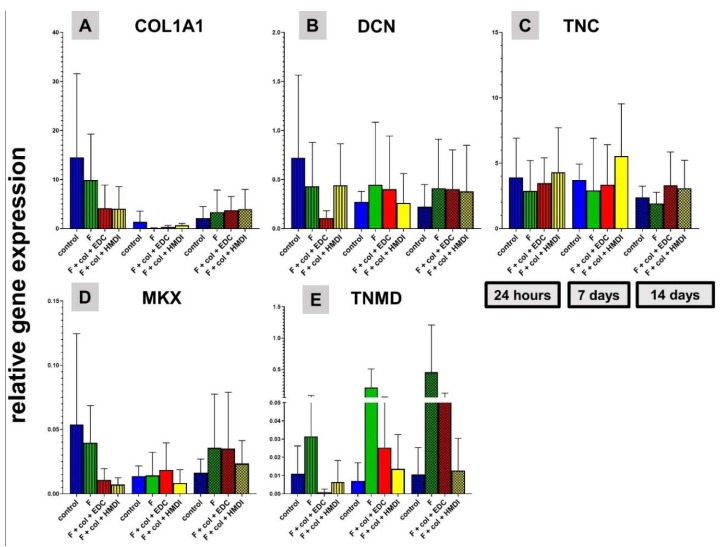
Expression level of genes for ligament extracellular matrix components and the transcription factor Mohawk (MKX) after 24 h, 7 days and 14 days in the different scaffold functionalization variants. Relative gene expression of type I collagen (COL1A1, **A**), of decorin (DCN, **B**), of tenascin C (TNC, **C**), Mohawk (MKX, **D**) and tenomodulin (TNMD, **E**). All diagrams summarize *n* = 5 independent experiments with cells of 5 different donors and show means with standard deviation. No significant difference could be calculated. Gene expressions were normalized to the reference gene GAPDH. Col, collagen foam; F, fluorinated; EDC, ethylcarbodiimide cross-linked; HMDI, hexamethylene diisocyanate cross-linked.

**Table 1 ijms-21-01132-t001:** Primers used in this study.

Gene Symbol	Species	Gene Name	Amplicon Length	Assay ID
*COL1A1*	*O. cuniculus*	collagen, type I, alpha 1	70	Oc03396073_g1
*DCN*	*Homo sapiens*	decorin	77	Hs00370384_m1
*TNC*	*O. cuniculus*	tenascin C	61	Oc06726696_m1
*MKX*	*O. cuniculus*	Mohawk	60	Oc06754037_m1
*TNMD (LOC100125994)*	*O. cuniculus*	mydulin	146	Oc03399505_m1
*GAPDH (LOC100009074)*	*O. cuniculus*	glyceraldehyde-3-phosphate dehydrogenase	82	Oc03823402_g1

O: Oryctolagus

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
