# Peer review of "Enhanced Growth of Lapine Anterior Cruciate Ligament-Derived Fibroblasts on Scaffolds Embroidered from Poly(l-lactide-co-ε-caprolactone) and Polylactic Acid Threads Functionalized by Fluorination and Hexamethylene Diisocyanate Cross-Linked Collagen Foams"

_ijms, 2020, doi:10.3390/ijms21031132_

Round 1
Reviewer 1 Report
The presented data submitted by Clemens et al. are novel and the manuscript fits the Special issue of journal; therefore this manuscript is suitable for publication in ijmc but it needs some minor formal revision
Minor:
Please use consistent form for some terms, for example use a space vor and after “=” (L 276 or L 310 …)
L 172, Please add „. “ After (Fig. 3I, J)
L 202 Please delet “.”
Author Response
Reviewer 1
The presented data submitted by Clemens et al. are novel and the manuscript fits the Special issue of journal; therefore this manuscript is suitable for publication in ijmc but it needs some minor formal revision
Minor:
Please use consistent form for some terms, for example use a space vor and after “=” (L 276 or L 310 …)
Response: We inserted blanks and added a point at the end of the respective sentence.
L 172, Please add „. “ After (Fig. 3I, J)
Response: A point was added at the end of the sentence.
L 202 Please delete “.”
Response: Done.

Reviewer 2 Report
The objective of this paper is the assessment of an embroidered scaffold from P(LA-CL)/PLA regarding the toxicity, proliferation, gene expression and morphology of lapine ACL derived fibroblasts. The scaffold was tested under different functionalization conditions, by fluorination, by two different crosslinking conditions of a collagen foam and with two different seeding conditions (statical and dynamical). All of them for tissue engineering of ACL.
The Introduction is correct, the references are suitable, and the results of the experimental work are well described.
But in my opinion, the work is weak regarding some point:
The scaffold is not described on this paper. It’s true that the reference 60 describes the process to manufacture the scaffold, but the preparation of the scaffold is crucial to understand the paper, and some explanation and details should be given, better than only a picture on the graphical abstract (fig. 1). On paragraph 4.1 is only repeated a paragraph existing on reference 13 (this reference is incomplete), but nothing is said about the reference 60, where the scaffold is described. Therefore, this reference and an explanation should be included on paragraph 4.1.On the other hand, the scaffold looks an embroidered fabric, but without any specific porosity, and this fact could produce a regeneration of the ligament only in the surface of the scaffold, leading to a week regenerated tissue. Moreover, even in case the cells could penetrate in the interior of the scaffold, these cells could be squashed when the scaffold was stretched. In my opinion, a porous scaffold can avoid this problem, and there are in the literature some paper pointing out in this direction.
Another aspect of this paper is the low significance of the results, but the results of experiments can’t be changed. The found differences between the different scaffolds are only statistically significant in a few essays. It’s true that it is said several times thorough the paper, but it reduces the conclusions of the paper. Anyway, the authors are honest and in the conclusion they only cite the significant results.
Related to the discussion of results, this section should be reduced and clarified, by summarizing more clearly the more important points. The paragraph between lines 341 and 355 could be moved to the introduction. It looks more appropriate in the introduction than in the discussion of results. Moreover, some paragraph repeat results already cited in the results (lines 359 to 365).
The PLA is a well known adherent material for the fibroblasts, in part due to its hydrophobicity. The authors should modify or support with a reference the sentence on line 68: “A disadvantage is that PLA alone provides only limited cell adhesion”.
What is the reason to carry out the cytotoxicity test with L929 cells? If all the other tests have been carried out with LACL- derived fibroblasts, why not the cell viability test?
On the caption to figure 5 should be indicated that these results correspond to dynamically seeded cells.
Author Response
Reviewer 2
The objective of this paper is the assessment of an embroidered scaffold from P(LA-CL)/PLA regarding the toxicity, proliferation, gene expression and morphology of lapine ACL derived fibroblasts. The scaffold was tested under different functionalization conditions, by fluorination, by two different crosslinking conditions of a collagen foam and with two different seeding conditions (statical and dynamical). All of them for tissue engineering of ACL.
The introduction is correct, the references are suitable, and the results of the experimental work are well described.
But in my opinion, the work is weak regarding some point:
The scaffold is not described on this paper. It’s true that the reference 60 describes the process to manufacture the scaffold, but the preparation of the scaffold is crucial to understand the paper, and some explanation and details should be given, better than only a picture on the graphical abstract (fig. 1).
Response: We added substantial more information in regard to the embroidering of the scaffold e.g. details of pattern and porosity (see 4.1.).
On paragraph 4.1 is only repeated a paragraph existing on reference 13 (this reference is incomplete), but nothing is said about the reference 60, where the scaffold is described. Therefore, this reference and an explanation should be included on paragraph 4.1.
Response: We have revised the section 4.1 and inserted the corresponding references.
On the other hand, the scaffold looks an embroidered fabric, but without any specific porosity, and this fact could produce a regeneration of the ligament only in the surface of the scaffold, leading to a week regenerated tissue. Moreover, even in case the cells could penetrate in the interior of the scaffold, these cells could be squashed when the scaffold was stretched. In my opinion, a porous scaffold can avoid this problem, and there are in the literature some paper pointing out in this direction.
Response: The scaffolds are produced in a special process similar to the lace embroidery. By washing out the water soluble base material, a continuous porous structure is obtained. The process and the evolving porosities are described here:
Breier, A. C. (2015): 2 - Embroidery technology for hard-tissue scaffolds A2 - Blair, Todd. In: T1 - 2 - Embroidery technology for hard-tissue scaffolds A2 - Blair, Todd (Hg.): Biomedical Textiles for Orthopaedic and Surgical Applications: Woodhead Publishing Series in Biomaterials: Woodhead Publishing, S. 23–43.
The porosity of the ACL-scaffolds was determined by µCT as described in [Breier 2015] and is in a range of 65-75 % (continuously through the whole structure). This porosity might impair the risk of squashing the cells during tension. However, running experiments using a mechanostimulator (under uniaxial tension) showed a high viability of the cells and homogenous cell distribution on the scaffolds after 2 d stimulation. However, when we section the scaffold to get view on a complete cross-section of the scaffold cells might be damaged.
To prove that the cells are not restricted solely to the surface of the scaffold we analyzed scaffold vertical cross-sections for cell penetration into inner parts of the scaffold by mean of DAPI cell nuclei staining. The result (7 days dynamical scaffold culture) revealed substantial cell penetration into inner scaffold parts, which was highest in the HMDI cross-linked variant (around 45% of the scaffold cross-sectional diameter). The cell penetration was significantly higher in functionalized compared to non-functionalized scaffold variants. Hence, the collagen foam did not restrict exploration of the scaffold by the cells.
Another aspect of this paper is the low significance of the results, but the results of experiments can’t be changed. The found differences between the different scaffolds are only statistically significant in a few essays. It’s true that it is said several times thorough the paper, but it reduces the conclusions of the paper. Anyway, the authors are honest and in the conclusion they only cite the significant results.
Response: Using primary cells derived from at least three different donor animals in independent experiments leads generally to a higher degree of variation. The difference between the standard deviation in primary cells directly compared to that in a uniform fibroblast cell line (L929 fibroblasts) is visible now in the novel figure 2. The added measurements of cell penetration into the scaffold show a novel set with significant differences between scaffold variants.
Related to the discussion of results, this section should be reduced and clarified, by summarizing more clearly the more important points. The paragraph between lines 341 and 355 could be moved to the introduction. It looks more appropriate in the introduction than in the discussion of results. Moreover, some paragraph repeat results already cited in the results (lines 359 to 365).
Response: We removed redundant description of results in the discussion section and inserted the discussion of the novel data now.
The PLA is a well-known adherent material for the fibroblasts, in part due to its hydrophobicity. The authors should modify or support with a reference the sentence on line 68: “A disadvantage is that PLA alone provides only limited cell adhesion”.
Response: We modified the text in lines 68-72 accordingly and added references: „PLA has been described as a degradable, biomechanically robust, highly biocompatible and well processable polymer (Singvhi et al., 2019). However, a disadvantage is that PLA alone provides only limited cell adhesion as described for ACL fibroblasts previously (Hahn et al., 2019). Major influencing factors are its hydrophobicity (Singvhi et al., 2019) and the release of acidic degradation products (Zhang et al., 2016).“ Taken together, PLA has a proven biocompatibility and therefore a wide variety of applications, such as bone screws and pins for fixation, sutures, and drug delivery scaffolds. Nevertheless, the degradation of the polymer leads to acidic products, monomeric LA (lactic acid) or oligomers of LA. The cleaved monomers will diffuse out of the polymer over time and can change the pH in the immediate vicinity of the fibroblasts. Xu et al. 2011 demonstrated that in physiological pH of 7.4 – as it was also used in the study - PLA brushes have a degradation time of 100 h, while in acidic pH of 3 there was no verifiable degradation after 400 h. They also showed that PLA degraded 4 times faster in 37 °C in comparison to 25 °C.
What is the reason to carry out the cytotoxicity test with L929 cells? If all the other tests have been carried out with LACL- derived fibroblasts, why not the cell viability test?
Response: We performed cytotoxicity testing initially with L929 fibroblasts which are recommended by the guideline for cytotoxicity testing (International Standard DIN ISO 10993-5:2009). However, we added now the same testing procedure performed with lapine anterior cruciate derived fibroblasts (LACL-derived fibroblast). The results were very similar except for higher standard deviations, which could be explained by the fact that we included LACL-derived fibroblast from three different animals. Novel results are depicted in figure 2 now.
On the caption to figure 5 should be indicated that these results correspond to dynamically seeded cells.
Response: This information was added now.

Round 2
Reviewer 1 Report
The manuscript is improved and in my mind is sutible to publish by IJMS
Reviewer 2 Report
After the revision, I feel that this paper can be accepted fot publication in IJMS